# Analysis of the Circulating Tumor Cell Capture Ability of a Slit Filter-Based Method in Comparison to a Selection-Free Method in Multiple Cancer Types

**DOI:** 10.3390/ijms21239031

**Published:** 2020-11-27

**Authors:** Hidenori Takagi, Liang Dong, Morgan D. Kuczler, Kara Lombardo, Mitsuharu Hirai, Sarah R. Amend, Kenneth J. Pienta

**Affiliations:** 1Research and Development Division, ARKRAY, Inc. Yousuien-nai, 59 Gansuin-cho, Kamigyo-ku, Kyoto 602-0008, Japan; hiraim@arkray.co.jp; 2The James Buchanan Brady Urological Institute, Johns Hopkins University School of Medicine, 600 N. Wolfe Street, Baltimore, MD 21287, USA; ldong4@jhmi.edu (L.D.); mkuczle1@jhu.edu (M.D.K.); klombardo@jhmi.edu (K.L.); samend2@jhmi.edu (S.R.A.); kpienta1@jhmi.edu (K.J.P.); 3Department of Urology and Renji Hospital, Shanghai Jiao Tong University School of Medicine, 1630 Dongfang Road, Shanghai 200025, China

**Keywords:** circulating tumor cells, size-based filter, selection-free platform, prostate cancer, bladder cancer, kidney cancer, pancreatic cancer

## Abstract

Circulating tumor cells (CTCs) are a promising biomarker for cancer liquid biopsy. To evaluate the CTC capture bias and detection capability of the slit filter-based CTC isolation platform (CTC-FIND), we prospectively compared it head to head to a selection-free platform (AccuCyte^®^-CyteFinder^®^ system). We used the two methods to determine the CTC counts, CTC positive rates, CTC size distributions, and CTC phenotypes in 36 patients with metastatic cancer. Between the two methods, the median CTC counts were not significantly different and the total counts were correlated (*r* = 0.63, *p* < 0.0001). The CTC positive rate by CTC-FIND was significantly higher than that by AccuCyte^®^-CyteFinder^®^ system (91.7% vs. 66.7%, *p* < 0.05). The median diameter of CTCs collected by CTC-FIND was significantly larger (13.0 μm, range 5.2–52.0 vs. 10.4 μm, range 5.2–44.2, *p* < 0.0001). The distributions of CTC phenotypes (CK+EpCAM+, CK+EpCAM− or CK−EpCAM+) detected by both methods were similar. These results suggested that CTC-FIND can detect more CTC-positive cases but with a bias toward large size of CTCs.

## 1. Introduction

Circulating tumor cells (CTCs) are shed from primary and metastatic tumors into peripheral blood. CTCs are involved in cancer metastasis and recurrence [1]. Research on CTCs may lead to improved disease management, including monitoring, treatment decision making, and risk stratification [2]. Currently, the most commonly used device for analyzing CTCs is the CellSearch^®^ system (Menarini, Florence, Italy), which is the only Food and Drug Administration-approved assay for CTC enumeration. The system uses epithelial cell adhesion molecule (EpCAM) to capture the EpCAM-positive (EpCAM+) cells from peripheral blood followed by detection with cytokeratin (CK) markers and cluster of designation 45 (CD45) for distinguishing between CTCs and leukocytes. EpCAM+CK+CD45− cells are defined as CTCs [3]. This method has been used to assess CTC numbers for the prediction of treatment outcomes and progression-free survival in patients with metastatic breast, colorectal, and prostate cancers [4,5,6]. However, the existence of EpCAM-low expression/negative cells has been reported [7,8,9]. These cells may be of particular interest given that EpCAM is known to be downregulated during epithelial-to-mesenchymal transition, a process that is thought to be involved in metastasis [10,11,12]. Since such a critical subpopulation of CTCs cannot be isolated by EpCAM-positive selection. A system that collects CTCs independently of EpCAM is needed [13,14]. Thus, various alternative methods have been proposed for the detection and isolation/collection of CTCs from whole blood [15].

Filtering isolation is one CTC isolation method that does not depend on CTC surface protein markers. CTCs can be isolated from whole blood based on differences in cell size and deformability [16,17,18]. Erythrocytes and leukocytes have high deformability and they can pass through filters even when the pores of the filter are smaller than the size of the cells. Cancer cells are generally larger and have less deformability than erythrocytes and leukocytes, so they do not easily pass through filters with small pores. Various filters and filtering methods have been developed from the aspect of physical properties of target cells. A previous study reported cancer cell recovery rates of a filter with pore dimensions of 8 μm × 30 μm were higher than that of a filter with circle pores using a spike-in experiment [18]. Other studies reported that a parylene-C slot microfilter with pore dimensions of 6 μm × 40 μm and a hole-to-surface ratio of 18% was used for CTCs’ isolation [19]. They demonstrated that cell capture ability of the parylene-C slot microfilter was 90% capture efficiency, 90% cell viability, and 200-fold sample enrichment from 1 mL whole blood using a spike-in experiment. Filters with rectangular pores have reported high performance of cancer cells’ collection. However, little has been reported on analysis of CTC capture bias of filters with these pores.

In our previous study, we proposed a method of CTC enrichment by size-based filtration and immunomagnetic negative selection followed by dielectrophoretic concentration (CTC-FIND) [20]. The high purity and high recovery rate of this system were verified using a spike-in experiment. We used a high-precision metal filter with slit-shaped pore dimensions of 6.5 μm × 88 μm and a hole-to-surface ratio of 41%. Some CTCs were reported to be smaller than cultured cancer cells. The mean CTC size was reported to be 7.97 μm in prostate cancer [21]. Theoretically, this slit filter should enable the collection of CTCs with a cell diameter >6.5 μm. Recently, we developed an automated CTC-FIND system composed of a slit filter isolation unit and a purification and enrichment unit.

In this study, to analyze the CTC capture bias and detection capability of the slit filter isolation unit of CTC-FIND, we compared it to a selection-free method using the AccuCyte^®^-CyteFinder^®^ system (RareCyte, Inc., Seattle, WA, USA), which collects the buffy coat, including CTCs and leukocytes, from whole blood. The method of the AccuCyte^®^-CyteFinder^®^ system (the RareCyte method) can theoretically collect all nucleated cells, and the resulting population of CTCs is expected to be comprehensive [22]. One of the limitations in comparing two CTC technologies is that the disparities in protocols (i.e., different antibodies for CTC staining, different platforms for CTC visualization, etc.) may hide the differences caused by their main principle [23,24]. In order to decrease these biases in the present study, we modified the protocol for CTC-FIND to make it as comparable to AccuCyte^®^-CyteFinder^®^ system as possible (the CTC-FIND method). Specifically, all the cells enumerated by either platform were stained using the same reagents (RareCyte reagents) and were visualized and analyzed by the same instrument (CyteFinder^®^). In this case, any difference in CTC detection was most likely due to the slit-filtration vs. selection-free strategy. The CTC numbers and positive rates were compared between the CTC-FIND method and the RareCyte method. Additionally, to analyze the CTC capture bias of the slit filter-based CTC isolation method, we demonstrated differences in the size distributions of CTCs and the CTC phenotypes.

## 2. Results

### 2.1. Representative Images of CTCs Obtained by the Two Methods and Characteristics of the Methods

CTCs were analyzed in peripheral blood samples from metastatic cancer patients: 21 prostate cancer, nine bladder cancer, three kidney cancer, and three pancreatic cancer blood samples were collected for evaluating capture ability of the CTC-FIND method comparing to the RareCyte method. Representative images of CTCs are shown in Figure 1. Table 1 shows the differences in the two CTC detection methods.

### 2.2. Comparison of the Enumerated CTC Numbers between the Two Methods

To evaluate the CTC collection performance of the CTC-FIND method, a comparison analysis was performed with the RareCyte method. Overall, the median CTC count in 7.5 mL of whole blood was 9.0 CTCs (total of 934 CTCs, range 0–324) for the CTC-FIND method and 3.0 CTCs (total of 1046 CTCs, range 0–257) for the RareCyte method. The median CTC count in 7.5 mL of whole blood were not significantly different between the two methods (Wilcoxon matched-pairs signed-rank test, *p* = 0.0896, Figure 2a). A correlation was observed between the CTC counts obtained by the CTC-FIND method and the RareCyte method (Spearman’s rank correlation coefficient, *rs* = 0.63, *p* < 0.0001, Figure 2b). With the CTC-FIND method, the median CTC count in 7.5 mL of blood for each cancer type was as follows: prostate cancer, 9.0 CTCs (total of 776, range 0–324); bladder cancer, 10.0 CTCs (total of 106, range 2–23); kidney cancer, 13.0 CTCs (total of 48, range 3–32); and pancreatic cancer, 2.0 CTCs (total of 4, range 0–2). The corresponding values from the RareCyte method were as follows: prostate cancer, 4.0 CTCs (total of 978, range 0–257); bladder cancer, 2.0 CTCs (total of 40, range 0–16); kidney cancer, 5.0 CTCs (total of 23, range 4–14); and pancreatic cancer, 1.0 CTC (total of 5, range 1–3). For only bladder cancer, the median CTC count in 7.5 mL of blood differed significantly between the two methods (Wilcoxon matched-pairs signed-rank test, *p* = 0.0352, Figure 2c). In both the overall comparison and the comparisons in each cancer type, the median CTC counts tended to be higher with the CTC-FIND method than the RareCyte method.

### 2.3. Comparison of the Detected CTC Positive Rates between the Two Methods

To evaluate the performance of the CTC-FIND method in detecting CTC-positive samples, the percentage of CTC-positive samples was compared between the CTC-FIND method and the RareCyte method. CTC-positive samples were defined as those with ≥2 CTCs/7.5 mL of whole blood [24]. The concordance for the detection of positive samples between the CTC-FIND method and the RareCyte method was 66.7% (24/36, Figure 3a). For the overall analysis, the percentage of CTC-positive samples was 91.7% (33/36) by the CTC-FIND method and 66.7% (24/36) by the RareCyte method (Fisher’s exact test, *p* = 0.0182, Figure 3b). By the CTC-FIND method, the percentages of CTC-positive samples for each of the different cancer types were: prostate cancer, 95.4% (19/21); bladder cancer, 100% (9/9); kidney cancer, 100% (3/3); and pancreatic cancer, 66.7% (2/3). The corresponding values by the RareCyte method were: prostate cancer, 66.7% (14/21); bladder cancer, 66.7% (6/9); kidney cancer, 100% (3/3); and pancreatic cancer, 33.3% (1/3). The CTC-FIND method had a 25% higher percentage of positive samples than the RareCyte method in the overall comparison. Regarding the comparisons of separate cancer types, no significant difference in the percentage of CTC-positive samples was found (Figure 3c). The percentage of CTC-positive samples tended to be higher with the CTC-FIND method than with the RareCyte method in each cancer type.

### 2.4. Comparison of the CTC Sizes between the Two Methods

The sizes of CTCs that were collected by each method were measured by CyteFinder^®^ and compared. Figure 4a,b shows representative images of fields that contain a small CTC and a large CTC. The median diameter of CTCs obtained by the two methods was different: 13.0 μm (range, 5.2–52.0) from CTC-FIND and 10.4 μm (range, 5.2–44.2) from RareCyte (Mann Whitney test, *p* < 0.0001, Figure 4c). With the CTC-FIND method, the median diameter of CTCs in each cancer type were: prostate cancer, 13.0 μm (range, 5.2–39.0); bladder cancer, 11.7 μm (range, 5.2–52.0); kidney cancer, 13.0 μm (range, 7.8–37.7); and pancreatic cancer, 13.0 μm (range, 10.4–16.9). The corresponding values from the RareCyte method were: prostate cancer, 10.4 μm (range, 5.2–44.2); bladder cancer, 13.0 μm (range, 6.5–29.9); kidney cancer, 9.1 μm (range, 5.2–24.7); and pancreatic cancer, 9.1 μm (range, 7.8–15.6). In the overall analysis, the median size of the CTCs collected by the CTC-FIND method was larger than that of the CTCs collected by the RareCyte method. In each cancer type, the median diameters of the CTCs collected by the CTC-FIND method tended to be larger than those collected by the RareCyte method. For prostate cancer and kidney cancer, the median diameters of the CTCs collected by the CTC-FIND method were significantly larger than those collected by the RareCyte method (Mann Whitney test, prostate *p* < 0.0001, kidney *p* < 0.0001, Figure 4d). Figure 4e shows the CTC size distribution of the 934 CTCs collected via the CTC-FIND method and the 1046 CTCs collected via the RareCyte method. The 5th–95th percentile range for CTCs obtained by the CTC-FIND method was 7.8–22.1 μm, and that by the RareCyte method was 7.8–15.6 μm. With the CTC-FIND method, 2.6% of the CTCs had diameters of less than or equal to minor axis of the slit-filter pore (6.5 μm), whereas 4.2% of the CTCs collected with the RareCyte method did. Regarding the size distribution of the CTCs processed from the 36 samples, the CTC-FIND method collected many CTCs that were larger than or equal to 11.7 μm in diameter, while the RareCyte method collected many CTCs that were smaller than 11.7 μm. Thus, different CTC size distributions were obtained with the CTC-FIND method and the RareCyte method.

### 2.5. CTC Phenotype Analysis

An investigation was performed to determine whether the phenotype distribution of the CTCs collected by the CTC-FIND method and those collected by the RareCyte method differed based on cancer type. The CTC phenotypes were analyzed based on marker expression profiles. Figure 5a–c shows representative images of CTCs positive for CK and negative for EpCAM (CK+EpCAM− CTCs), negative for CK and positive for EpCAM (CK−EpCAM+ CTCs), and positive for both markers (CK+EpCAM+ CTCs). Next, we investigated the CTC phenotypes for each cancer type. The percentages of each phenotype by cancer type are shown in Figure 5d,e. Differences in the distribution of phenotypes were observed depending on the cancer type. Prostate CTCs tended to have higher rates of CK+EpCAM+ CTCs when compared to the CTCs of the other cancer types in both methods (for the CTC-FIND method: prostate cancer 62.8% vs. bladder cancer 9.4%, *p* < 0.0001; prostate cancer 62.8% vs. kidney cancer 20.8%, *p* < 0.0001; prostate cancer 62.8% vs. pancreatic cancer 0%, *p* = 0.0197; for the RareCyte method: prostate cancer 77.1% vs. bladder cancer 17.5%, *p* < 0.0001; prostate cancer 77.1% vs. kidney cancer 17.4%, *p* < 0.0001; prostate cancer 77.1% vs. pancreatic cancer 40%, not significant (ns), *p* > 0.05; Fisher’s exact test for each). Bladder CTCs and kidney CTCs had higher rates of CK+EpCAM− CTCs when compared to prostate cancer CTCs in both methods (for the CTC-FIND method: bladder cancer 85.0% vs. prostate cancer 30.0%, *p* < 0.0001; kidney cancer 77.1% vs. prostate cancer 30.0%, *p* < 0.0001; bladder cancer 85.0% or kidney cancer 77.1% vs. pancreatic cancer 75%, ns, *p* > 0.05; bladder cancer 85.0% vs. kidney cancer 77.1%, ns, *p* > 0.05; for the RareCyte method: bladder cancer 82.5% vs. prostate cancer 21.2%, *p* < 0.0001; kidney cancer 82.6% vs. prostate cancer 21.2%, *p* < 0.01; bladder cancer 82.5% or kidney cancer 82.6% vs. pancreatic cancer 40%, ns, *p* > 0.05; bladder cancer 82.5% vs. kidney cancer 82.6%, ns, *p* > 0.05; Fisher’s exact test for each). Both methods had captured the trends of the main CTC phenotypes of each cancer type.

## 3. Discussion

To evaluate the CTC collection performance and bias of the slit filter-based CTC isolation method, we compared it to a selection-free method (the RareCyte method). We investigated 36 blood samples of multiple cancer types for the CTC counts, CTC positive rates, CTC size distributions, and CTC phenotypes.

A previous study reported issues in comparisons, notably a potential lack of uniformity in the definition of CTCs and methodological biases [23]. Methodological biases include collection bias, detection bias, and analysis bias. Regarding comparative research, our research aimed to properly evaluate the collection performance of the filtration unit and its collection biases. The same antibody reagents and the same CTC imaging/analysis instrument were used for both methods to reduce the CTC detection bias and analysis bias. This study design provided greater uniformity between the two methods and reduced many potential biases, including differences in the composition of antibody reagents, in image acquisition methods (e.g., resolution and sensitivity of the camera), and in the algorithms used to determine the CTC counts. This study design helped to understand accurately the CTC collection performance and bias through the comparison of the CTC-FIND method and the RareCyte method.

The RareCyte method is a selection-free method based on the idea of “no cell left behind”. Theoretically, all nucleated cells should be deposited onto the target microscope slide [25]. Therefore, the RareCyte method is expected to offer a comprehensive CTC collection. In contrast, the filter-based system could isolate CTCs from whole blood, but some loss of CTCs can occur. Nonetheless, our results indicated that the CTC-FIND method and the RareCyte method did not differ significantly in the median CTC count, and the data from the two methods showed a correlation (Figure 2a,b). Additionally, our data indicated that the CTC-FIND method detected more CTC-positive blood samples than did the RareCyte method (Figure 3b). The Result of CTC-positive blood sample rates by the CTC-FIND method may be related to the tendency that the median CTC counts were higher with the CTC-FIND method than the RareCyte method. In the RareCyte method, in actual practice, there are multiple steps that can cause CTC loss, i.e., pipetting, tube switching, etc. Particularly, in the RareCyte system, a certain proportion of CTC loss could happen during the staining process on the slides. Instead, in the CTC-FIND system, CTCs were stained directly on the filter, which may prevent the CTC loss during staining since they could not pass through the filter pores. These differences of such processes might make CTC counts of the CTC-FIND method higher than that of the RareCyte method.

The median sizes of the CTCs collected by the CTC-FIND method were significantly larger than those collected by the RareCyte method (Figure 4c,d). To investigate why the median size of the collected CTCs differed between the two methods, we examined the distribution of the CTC sizes in the two methods (Figure 4e). The 5th to 95th percentile range of the CTC sizes differed between the CTC-FIND method and the RareCyte method: The range was 6.5 μm larger in the CTC-FIND method than in the RareCyte method (7.8–22.1 μm vs. 7.8–15.6 μm, respectively). The size distribution of CTCs yielded by each method suggested that the two methods have different characteristic shapes of the CTC size distribution. It is possible that large CTCs are overrepresented in the CTC-FIND method due to filter capture. In the RareCyte method, it is possible that large CTCs on the slide might have been washed away during the staining processing. There are multiple washing steps before or within the Auto-Stainer. As result of these processes, large CTCs on the slide might be easily washed away compared to small cells during the staining processing. This is because shear stress caused by the liquid flow affects large cells more strongly than small cells [26]. Another possibility is that large CTCs may be lost when the density of a large CTC (the specific gravity of a large CTC) is slightly higher and the large CTCs may move to an erythrocyte layer in the AccuCyte tube after using a centrifuge. Next, the difference in the protocols may also cause some artifacts on cell size, i.e., the difference in a collection tube, storage period, and staining process between two methods. For instance, an AccuCyte blood collection tube makes CTCs stable for 72 h because the tube contains fixation reagent for CTCs. Prefixation of cells in the blood collection tube can cause some level of cell shrinkage. According to the article using the similar concept of the blood collection tube [21], it has been reported ~6% cell shrinkage during 48 h preservation when using Cellsave™ (Menarini Silicon Biosystems, Castel Maggiore, ITALY), which includes fixation reagent. The fixation reagent also makes cells rigid (loss of deformability). These cells will clog the narrow slit-shaped filter. For this reason, we did not use the AccuCyte blood collection tube for CTC-FIND. Generally, since the EDTA tube is used for blood morphology, we selected this tube for CTC-FIND. However, the cells staying in EDTA for a long time might affect the cell size. These differences (the differences in the collection tubes, storage period, staining process) will need to be validated.

Regarding the filter-based isolation method, cells that are smaller than the pores would be missed. The present study employed a slit filter with a pore size of 6.5 μm (minor axis) × 88 μm (major axis) for collecting CTCs. Theoretically, this slit filter should enable the collection of CTCs with a cell diameter >6.5 μm. Indeed, CTCs larger than 6.5 μm in diameter were collected by the CTC-FIND method. From a different perspective, among all CTCs collected by the RareCyte method, only 4.2% were CTCs smaller than or equal to 6.5 μm in diameter. These data indicated that the setting used for the filter pore width was appropriate for collecting CTCs. However, the count of small CTCs (< 11.7 μm) was lower in the CTC-FIND method than in the RareCyte method (Figure 4e).

Next, in our investigation, we noticed that the values of three prostate cancer samples (out of the total 36 blood samples) greatly affected the total CTC count. In those three prostate cancer samples (the three PCa), the RareCyte method was able to collect several hundred CTCs, while the CTC-FIND method collected less than half that number of CTCs. We hypothesized that the reason for this was that the blood samples with a significantly higher CTC count by the RareCyte method than the CTC-FIND method were blood samples containing a large number of small CTCs (< 11.7 μm).

In the three PCa cases, the CTC-FIND method had a substantially lower CTC count than the RareCyte method regardless of CTC size (Appendix A). It is possible that this is due to an increase in filtration pressure. A previous study reported that a high filtration pressure led to low CTCs’ recovery [27]. Due to the high number of CTCs in these samples, it is possible that the filter was clogged. In a previous study for a filter with rectangular pores, the number of leukocytes captured varied widely among patients compared to the number of leukocytes from normal blood samples [18]. In other possibilities, leukocytes may have become less deformable due to disease-related conditions [28]. It is possible that small aggregations may have occurred in the whole blood due to plasma protein concentration or hematocrit- or disease-related conditions [29,30,31,32]. Thus, these components may have easily clogged parts of the filter pores and/or increased the viscosity of the sample, resulting in a filtration pressure increase when a constant flow is used. As a result, CTCs may have passed through the filter regardless of the CTC size or may have been destroyed by edge of filter pores [33]. To improve the collection performance in similar cases, the removal of leukocytes and the hemolysis of erythrocytes can be performed before filtration, or other blood stabilization reagents can be performed [34,35,36,37]. Also, as a method of suppressing the increase in filtration pressure, when filtering blood, it is possible to use a constant pressure setting rather than a constant flow setting.

To investigate whether the major CTC phenotypes of the different cancer types differed between the two methods, the major CTC phenotypes were compared between the CTCs of the CTC-FIND method and the RareCyte method. In both methods, the major CTC phenotype of the prostate cancer blood samples was CK+EpCAM+. In bladder and kidney cancers, the major CTC phenotype was CK+EpCAM−. The trends of the major CTC phenotypes in each cancer type were similar between the CTCs of the CTC-FIND method and the RareCyte method (Figure 5d,e). In this study, the subtype of all the kidney cancers was clear cell renal cell carcinoma, which is a major subtype of kidney cancer. The low frequency of EpCAM+ CTCs in the blood samples from kidney cancer patients may be related to the fact that clear cell renal cell carcinoma tissue specimens show low levels of EpCAM [38,39]. We verified that the CTC-FIND method could collect CTCs with the major CTC phenotypic trends of different cancer types as well as the RareCyte method. Both methods demonstrated that CK−EPCAM+ CTCs were very rare/absent in bladder cancer and kidney cancer samples. Even in the CTC-FIND method, only 5.6% and 2.1% CTCs were CK−EPCAM+, which was 6 out of 106 and 1 out of 48. The difference between the two methods in this subtype was not statistically significant (Fisher’s exact test, *p* > 0.05). For pancreatic cancer, each phenotype rate between both methods seemed to be different. This difference can be due to the limited number of CTCs captured by either method in pancreatic cancer patients. From three pancreatic cancer blood samples, a total of four CTCs were yielded by the CTC-FIND method and a total of five CTCs were yielded via the RareCyte method.

In this study, prostate cancer contributed dominantly to overall results. These results may also be affected by the proportion of cases in each cancer type. These findings need to be further validated in other cancer types and a larger cohort size.

The slit-filter isolation unit of CTC-FIND could automatically provide cell suspension including CTCs. It is potentially versatile because the cell suspension can then be applicable to an instrument for various analyses. Furthermore, the use of the whole CTC-FIND system, which integrates a filtration unit with a purification and enrichment unit, offers the potential to collect high-quality CTCs. The number of blood samples included in this study was not large enough to explore any relationships among the number of CTCs, patient characteristics, or clinical significance. In the future, the clinical utility of CTC-FIND will be investigated by evaluating a larger number of samples.

## 4. Materials and Methods

### 4.1. Study Design

The study was performed to investigate the CTC capture bias and detection capability of the slit-filter unit of the CTC-FIND (Figure 6). To decrease methodological biases (detection biases and analysis biases), RareCyte reagents were used for immunostaining in both methods. The reaction time was also set to be the same. Additionally, the same imaging analysis instrument (CyteFinder^®^) was used for both methods. Table 1 shows the differences in the two CTC detection methods.

### 4.2. Sample Collection

With the consent of the patients, peripheral blood was collected from metastatic cancer patients: 21 prostate cancer, nine bladder cancer, three kidney cancer, and three pancreatic cancer blood samples were collected. The blood from each patient was taken during the same blood draw into two tubes: a vacuum blood collection tube containing ethylenediaminetetraacetic acid (EDTA) that was used for the CTC-FIND method and an AccuCyte^®^ blood collection tube (RareCyte, Inc., Seattle, WA, USA) that was used for the RareCyte method. The blood in the AccuCyte^®^ blood collection tube was stored at room temperature and 7.5 mL of blood was used for the RareCyte method within 48 h. The blood in the EDTA tube was stored at room temperature, and 7.5 mL of blood was used for the CTC-FIND method within 10 h.

### 4.3. CTCs Collected by the Slit Filter-Based Isolation of the CTC-FIND Method

CTC-FIND system is an automatic CTC collection system and composed of a slit-filter isolation unit and a purification and enrichment unit. After loading the reagents onto the system and adding blood, all subsequent steps were performed automatically. In this study, the slit-filter isolation unit of CTC-FIND was used. A filtration module of the slit-filter isolation unit was constructed by sandwiching the slit filter between the sample reservoir and the outlet connected to a tube pump [20].

The sample reservoir was filled with phosphate-buffered saline (PBS; Thermo Fisher Scientific, Waltham, MA, USA) from the lower side of the reservoir. PBS addition was stopped when the level was slightly above the filter. PBS was also used to fill the channel extending from the filter to the waste liquid bottle via tube pump. Whole blood (7.5 mL) was loaded onto the sample reservoir and filtered, which was then washed with PBS. The residual erythrocytes on the filter were hemolyzed using hemolysis reagent (BD, Franklin Lakes, NJ, USA) for 10 min at room temperature. The trapped cells, including CTCs and leukocytes, on the filter were then washed with PBS. The trapped cells were fixed by incubation with 10% neutral buffered formalin (Sigma-Aldrich, St. Louis, MO, USA) for 10 min at room temperature. The trapped cells were then washed with PBS, followed by incubation for 5 min with Tris-buffered saline (TBS, Quality Biological Inc, Gaithersburg, MD). Subsequently, the trapped cells were subjected to cell membrane permeation by incubation for 10 min in PBS containing 0.5% (*v*/*v*) Tween 20 (Thermo Fisher Scientific, Waltham, MA, USA) after the cells were washed with PBS. Next, after the trapped cells were washed with PBS again, the cells were incubated for 10 min at room temperature in blocking buffer consisting of PBS containing 0.2% (*w*/*v*) bovine serum albumin (Thermo Fisher Scientific Waltham, MA, USA) and fragment crystallizable receptor (FcR) blocking reagent, human (1:10, Miltenyi Biotec, Bergisch Gladbach, Nordrhein-Westfalen, Germany). Cell staining was performed using the same reagents used for the RareCyte method. Information on the cell staining reagents is listed in Section 4.6. The blood filtering speed was 200 μL/min; other liquid filtering speeds (e.g., washing buffer, fixing reagent, membrane permeation reagent, and staining reagent) were 1000 μL/min.

To collect the trapped cells from the filter after cell staining, in the final step of the slit-filter isolation unit of the CTC-FIND, the stained cells containing CTCs and leukocytes were collected by backflow with PBS containing 0.2% (*w*/*v*) bovine serum albumin. The cell suspension (approximately 800 μL) obtained from the filter was transferred into a 2-methacryloyloxyethyl phosphorylcholine (MPC) polymer-coated, 1.5 mL tube (SARSTEDT, Nümbrecht, Germany) and then manually spread onto two Superfrost^®^ Plus slides (VWR, Radnor, PA, USA; 400-μL each). The cell suspension on the slides was allowed to air dry. An aliquot (100 μL) of PBS was dripped onto each dried slide, and a coverslip (Thermo Scientific™ Richard-Allan Scientific™ Cover Glass, Thermo Fisher Scientific, Waltham, MA, USA) was placed on top of the spot. RareCyte mountant was used to prevent evaporation around the coverslip and slide sample. The mounted slide was subjected to centrifugation at 500× *g* for 5 min at room temperature.

### 4.4. CTCs Collected by AccuCyte^®^ System of the RareCyte Method

Whole blood (7.5 mL) in the AccuCyte^®^ blood collection tube was processed manually using an AccuCyte^®^ system (RareCyte, Inc., Seattle, WA, USA). The blood was used within 48 h of collection. Processing steps were performed according to the manufacturer’s protocol [40]. The collected sample (buffy coat) was spread onto each of eight Superfrost® Plus slides (VWR, Radnor, PA, USA) and air-dried at room temperature. The dried slides were stored at −20 °C before the subsequent staining procedure.

### 4.5. Slide Staining of the RareCyte Method (Pretreatment of Slide Specimens

The slides were stained according to the previously published protocol [40]. Briefly, the slides were allowed to equilibrate to room temperature before being fixed by immersion in 10% neutral buffered formalin solution (Sigma-Aldrich, St. Louis, MO, USA) for 60 min at room temperature. Next, TBS (Quality Biological Inc.) was used to wash the slides twice. These washes neutralized any remaining formalin. Before being stained in the AutoStainer Link 48 (Agilent Technologies, Santa Clara, CA, USA), the slides were subjected to antigen retrieval by incubation in a Tris-HCl buffer, pH 10 (Sigma-Aldrich, St. Louis, MO, USA), for 6 min at 75 °C. These slides were quenched in cool PBS for 5 min. This process improved the accessibility of the antigens to the antibodies in the staining reagents. After cell staining, RareCyte mountant was used with coverslips (Thermo Fisher Scientific, Waltham, MA, USA) on the slides, which were allowed to dry overnight at room temperature (25 °C) in the dark.

### 4.6. The Same Cell Staining Reagents and Reaction Time

The same cell staining reagents and the same reaction time were used for the CTC-FIND method and the RareCyte method. Specifically, cell staining was performed using a proprietary custom reagent kit developed by RareCyte. This included the following: 4′,6-diamidino-2-phenylindole (DAPI) to stain the cell nucleus) and antibodies specific for the epithelial marker CK, the epithelial marker EpCAM, and the cell surface antigen CD45/CD66b (used for staining leukocytes). We modified the assay by adding additional counter-stain reagent with the following phycoerythrin (PE)-conjugated antibodies: anti-CD14-PE (Clone M5E2, 1:200, BioLegend, San Diego, CA, USA), anti-CD34-PE (Clone 581, 1:200, BioLegend, San Diego, CA, USA), and anti CD11b-PE (Clone M1/70, 1:200, BioLegend, San Diego, CA, USA). For the CTC-FIND method, all staining was done with the slit-filter isolation unit of CTC-FIND. For the RareCyte method, all staining was done with the AutoStainer Link 48 (Agilent Technologies, Santa Clara, CA, USA) and according to manufacturer’s protocol.

### 4.7. Slide Scanning and the Size Measurement of Cell Images by the Same Imaging System (CyteFinder^®^)

For the RareCyte method, slides were scanned using CyteFinder^®^ (RareCyte, Inc., Seattle, WA, USA) according to the published protocol [40]. For the CTC-FIND method, the fluorescence exposure time and threshold were determined based on the results of a spike-in experiment. Each whole slide was scanned at a magnification of 10×. In this study, a CTC candidate was defined to be a CTC if the cell image listed by CyteFinder^®^ met the following criteria: a DAPI-positive nucleus with a diameter of 4 μm or more, positive for CK staining, and negative for the stained color comparisons of CD45, CD66b, CD11b, CD14, and CD34.

For the size measurement of cell images, CyteFinder^®^ software is able to display circles for the size measurement of cell images on a computer screen and to measure the sizes of CTCs by using the circles, which can change in circle diameter size by a pitch of 1.3 µm. The cell diameter for each CTC was defined as the size of the circle.

### 4.8. Statistical Analysis

The Wilcoxon matched-pair signed-rank test was used to compare CTC counts between the CTC-FIND method and the RareCyte method. Spearman’s rank correlation coefficient was used to analyze the correlation between the CTC-FIND method and the RareCyte method. Fisher’s exact test was used to compare CTC-positive sample rates between the CTC-FIND method and the RareCyte method and to compare CK+EpCAM+ CTC detection rates among cancer types. The Mann Whitney test was used to compare the median CTC diameters between the CTC-FIND method and the RareCyte method. *p* values < 0.05 were considered significant. All analytical statistics were performed using Prism (v 8; GraphPad, San Diego, CA, USA).

### 4.9. Ethical Statement

All subjects gave their informed consent for inclusion before they participated in the study. The study was conducted in accordance with the Declaration of Helsinki, and the protocol was approved by the Ethics Committee of Johns Hopkins Medicine Institutional Review Board (NA_00087094, May, 24, 2018).

## 5. Conclusions

By making the evaluation protocols more uniform in the comparison between CTC-FIND and the selection-free method (the RareCyte method), we could verify the CTC capture ability of the slit-filter isolation unit of CTC-FIND. Our data highlighted that CTC-FIND can detect more CTC-positive cases; however, it also had biases toward large size of CTCs. Further studies are needed on the clinical utility of CTC-FIND.

## Figures and Tables

**Figure 1 ijms-21-09031-f001:**
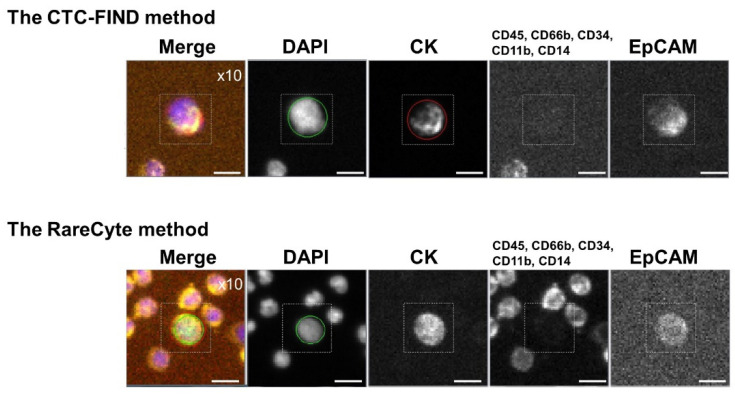
Representative images of circulating tumor cells (CTCs) collected by each method. Representative immunofluorescence images of CTCs detected by the slit filter-based CTC isolation unit of CTC-FIND and the CyteFinder^®^ system (the CTC-FIND method), and the AccuCyte^®^-CyteFinder^®^ system (the RareCyte method). CTCs were defined as cells with positive staining for cytokeratin (CK) and/or epithelial cell adhesion molecule (EpCAM), but negative for cluster of designation 45 (CD45), CD66b, CD34, CD11b, and CD14. The 4′,6-diamidino-2-phenylindole (DAPI) staining shows the presence of cells. The scale bars indicate 10 μm. Cells were scanned at a magnification of 10×. The excitation (Ex) and emission (Em) wavelengths for each biomarker were as follows: DAPI, Ex 390 nm, Em 431 nm; CK, Ex 475 nm, Em 525 nm; CD45, CD66b, CD34, CD11b, and CD14, Ex 542 nm, Em 590 nm; and EpCAM, Ex 632 nm, Em 690 nm.

**Figure 2 ijms-21-09031-f002:**
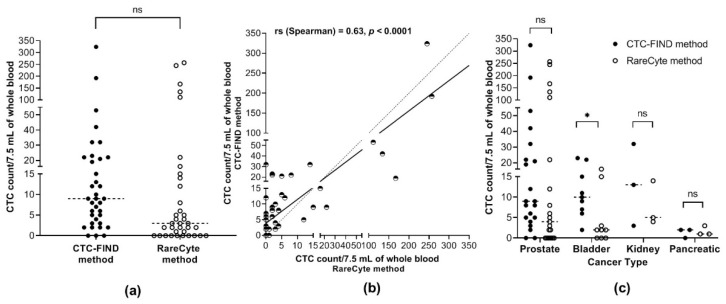
CTC enumeration. (**a**) The CTC counts in 7.5 mL of whole blood by each method. The CTC counts from each whole blood sample by the CTC-FIND method and the RareCyte method are indicated as ● (filled circles) and ○ (open circles), respectively. The CTC counts from 36 blood samples are shown. The median CTC count in 7.5 mL of whole blood was 9.0 and 3.0 for the CTC-FIND method and the RareCyte method, respectively (Wilcoxon matched-pairs signed-rank test, *p* = 0.0896). (**b**) Correlation in the total CTC counts between the two methods. The correlation analysis is shown by plotting the CTC counts in 7.5 mL of whole blood by the CTC-FIND method and the RareCyte method on the Y- and X-axes, respectively. These points are indicated as the half-empty circles. The correlation between the total CTC counts obtained by the CTC-FIND method and the RareCyte method was significant (Spearman’s rank correlation coefficient, *rs* = 0.63, *p* < 0.0001). An identity line is shown as a dashed line. Simple linear regression is shown as a solid line. (**c**) Each method was used to determine the CTC counts in 7.5 mL of whole blood in four different types of cancers. The CTC counts from each sample by the CTC-FIND method and the RareCyte method are indicated as ● (filled circles) and ○ (open circles), respectively. Wilcoxon matched-pairs signed-rank test was used. *: *p* < 0.05; ns: not significant.

**Figure 3 ijms-21-09031-f003:**
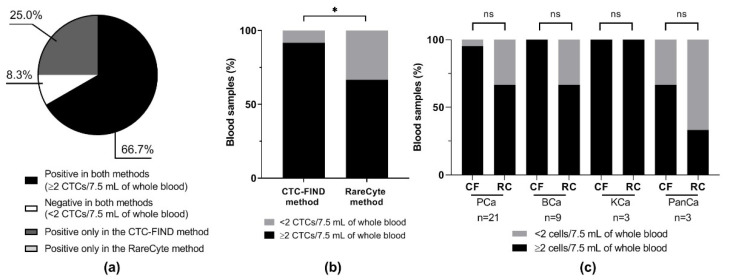
CTC positive rate. (**a**) Concordance between the two methods for the identification of CTC-positive and -negative samples. CTC-positive samples were defined as those with ≥2 CTCs/7.5 mL of blood. (**b**) The percentage of CTC-positive samples among all samples as assessed by the two methods. In this bar chart, black shading indicates the percentage of CTC-positive samples (out of 36 in total). The percentage of CTC-positive samples was 91.7% with the CTC-FIND method and 66.7% with the RareCyte method; they differed significantly (*p* = 0.0182 by Fisher’s exact test). (**c**) Frequencies of CTC-positive status for each cancer type. In this bar chart, black shading indicates the percentage of CTC-positive samples (defined as in (**a**))**.** CF: the CTC-FIND method; RC: the RareCyte method; PCa: prostate cancer; BCa: bladder cancer; KCa: kidney cancer; PanCa: pancreatic cancer; ns: not significant (*p* > 0.05 by Fisher’s exact test); *: *p* < 0.05.

**Figure 4 ijms-21-09031-f004:**
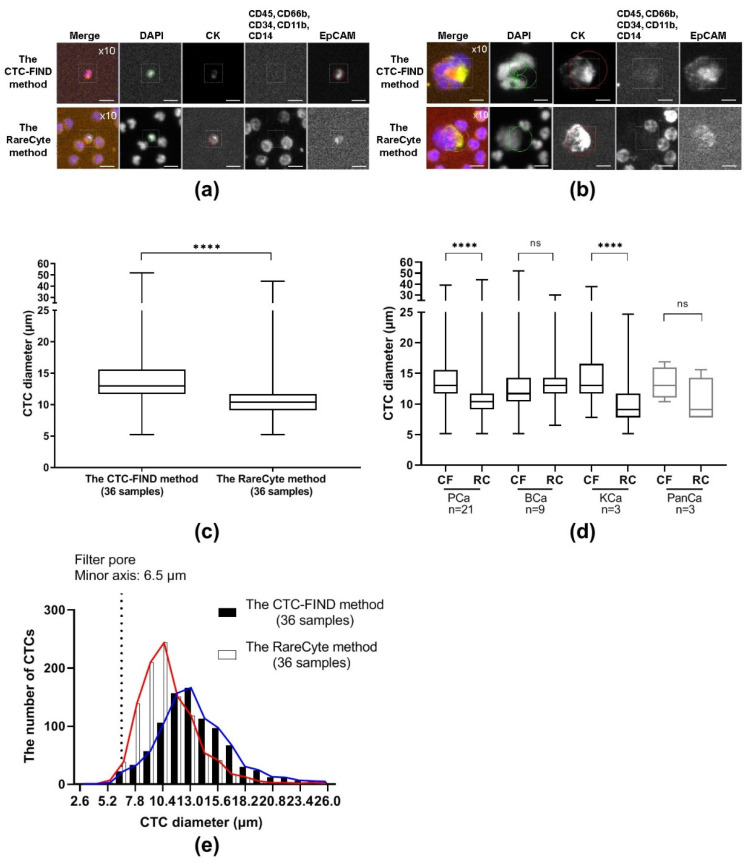
CTC diameter analysis. Representative immunofluorescence images of (**a**) small CTCs and (**b**) large CTCs detected by the CTC-FIND method and the RareCyte method. The scale bars indicate 10 μm. Cells were scanned at a magnification of 10×. Each staining image in a given row was taken from the same field of cells. (**c**) The diameters of each CTC were plotted as a box plot based on the median, interquartile ranges, and maximum and minimum whiskers for each method. The median CTC diameters obtained by the two methods were significantly different (13.0 µm for the CTC-FIND method and 10.4 µm for the RareCyte method; *p* < 0.0001 by Mann Whitney test). (**d**) The diameters of CTC in each cancer type were plotted as a box plot based on the median, interquartile ranges, and maximum and minimum whiskers for each method. (**e**) The CTC diameter distribution and the number of CTCs in each range from each of the two methods. The black and white bars indicate the CTCs obtained by the CTC-FIND method and the RareCyte method, respectively. The X-axis indicates the cell diameter classifications. The graph shows an X-axis with 2.6-μm intervals, i.e., the bar graph between 5.2 µm and 7.8 µm on the X-axis is at 6.5 µm. Blue and red lines indicate a connecting line of the CTC-FIND method and the RareCyte method, respectively. CF: the CTC-FIND method; RC: the RareCyte method; PCa: prostate cancer; BCa: bladder cancer; KCa: kidney cancer; PanCa: pancreatic cancer; ns: not significant (*p* > 0.05 by Mann Whitney test); ****: *p* < 0.0001

**Figure 5 ijms-21-09031-f005:**
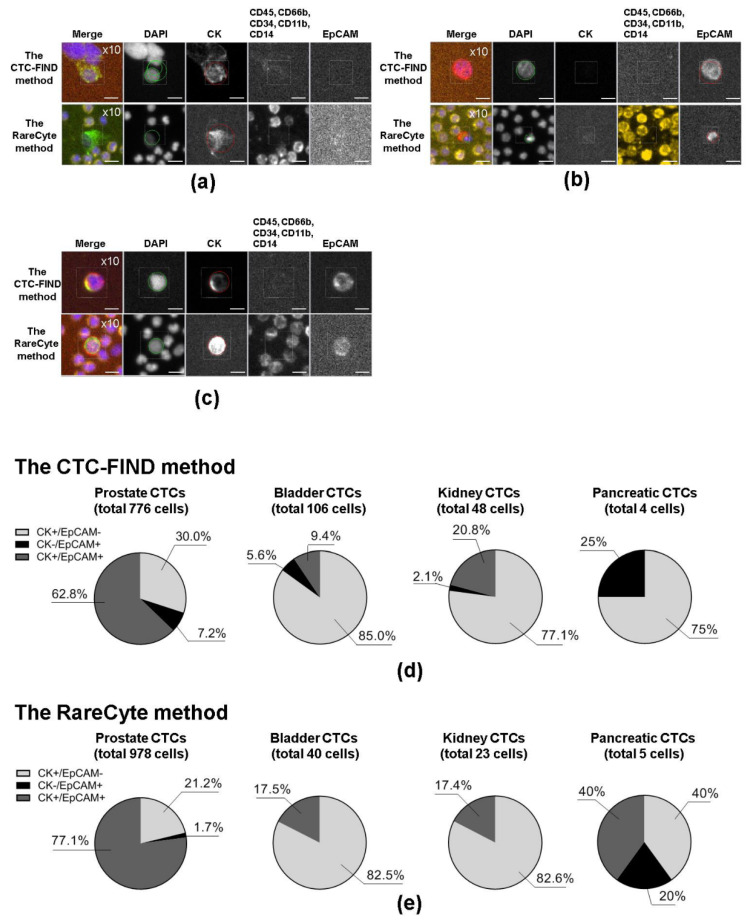
CTC phenotype analysis. Panels (**a**–**c**) show representative immunofluorescence images of CTCs with various marker phenotypes. The marker expression phenotypes shown are (**a**) CK+EpCAM−, (**b**) CK−EpCAM+, and (**c**) CK+EpCAM+. The scale bars indicate 10 μm. Cells were scanned at a magnification of 10×. Each staining image in a given row was taken from the same field of cells. (**d**) Population of CTCs with each phenotype among the CTCs detected by the CTC-FIND method in each cancer type. (**e**) Population of CTCs with each phenotype among the CTCs detected by the RareCyte method in each cancer type.

**Figure 6 ijms-21-09031-f006:**
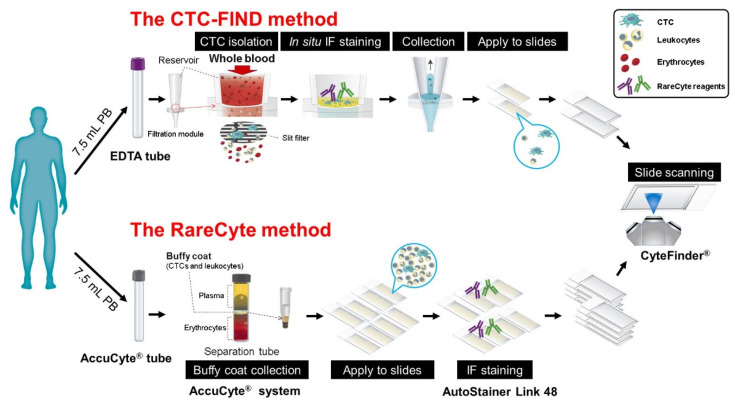
Schematic diagram of the study design for the counting of CTCs. A schematic diagram of the techniques used by the CTC-FIND method and the RareCyte method. Peripheral blood from each cancer patient was collected into two collection tubes: an ethylenediaminetetraacetic acid (EDTA) tube for the CTC-FIND method and an AccuCyte^®^ blood collection tube for the RareCyte method. The CTC-FIND method uses a slit-filter membrane for the isolation of CTCs, and CTCs were isolated from 7.5 mL of whole blood. After filtering, the trapped cells, including CTCs and leukocytes, were then subjected to in situ staining and collected into a 1.5 mL tube. Next, these cells were spread onto two positively charged slides, followed by the identification and enumeration of the CTCs using CyteFinder^®^. In the RareCyte method, the AccuCyte^®^ system was used to separate nucleated cells in the buffy coat, including the CTCs and leukocytes, from 7.5 mL of whole blood. The resulting population of nucleated cells was then spread onto eight positively charged slides and immunostained using an AutoStainer Link 48. The CTCs were scanned and identified by CyteFinder^®^ to enumerate them. CTCs were analyzed in 21 prostate cancer patient blood samples, nine bladder cancer patient blood samples, three kidney cancer patient blood samples, and three pancreatic cancer patient blood samples by using each method.

**Table 1 ijms-21-09031-t001:** Characteristics of the CTC isolation methods.

Assay Characteristics	The CTC-FIND Method	The RareCyte Method
CTC detection strategy	Physical property-based isolation (size and deformability)	Selection-free
* Markers used in the assay	DAPI, CK, EpCAM, CD45, CD66b, CD34, CD11b, CD14	DAPI, CK, EpCAM, CD45, CD66b, CD34, CD11b, CD14
Staining	On a filter	On a slide
Scanning/Size measurement	CyteFinder^®^	CyteFinder^®^

* DAPI: 4′,6-diamidino-2-phenylindole (nucleus marker); CD: cluster of designation; CD45, CD66b, CD34, CD11b, CD14: markers of lymphocytes, granulocytes, hematopoietic stem and progenitor cells, and endothelial cells; CK: cytokeratin; EpCAM: epithelial cell adhesion molecule. CK and EpCAM are epithelial markers used as markers of CTCs.

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
