# Peer review of "Analysis of the Circulating Tumor Cell Capture Ability of a Slit Filter-Based Method in Comparison to a Selection-Free Method in Multiple Cancer Types"

_ijms, 2020, doi:10.3390/ijms21239031_

Round 1

Reviewer 1 Report

This manuscript is eminently technical in nature, describing the comparison between a filter-based CTC isolation procedure (CTC-FIND method) and a theoretically unbiased selection procedure (RareCyte method) to determine CTCs in blood as a biomarker in cancer. Interesting methodological considerations are made on the two methods and the comparison between them is designed so that the possible differences detected are mainly due to the sample collection and enrichment stages. The main conclusions reached by the authors are that both methods detect similar amounts of CTCs, although the CTC-FIND method classifies as positive a higher proportion of diverse cancer patient derived samples according to the criterion of positivity >2 CTCs/7.5 blood mL. Additionally, the CTC-FIND method shows a bias towards the detection of larger CTCs although they carry a comparable surface markers distribution. These findings are of interest if the CTC-FIND method aims to be validated for a clinical use.

Some minor language issues should be corrected in the final edition work.

The methodology described is sound and the conclusions reached by the authors seem to be correct according to the presented data. However, some points could be addressed by the authors in order to offer a better understanding and interpretation points to potential readers.

Point 1:

According to figure 2b, for approximately two thirds of the samples the CTC-FIND counts are higher than RareCyte counts. How do the authors explain this if the RareCyte method as it is expected to be comprehensive in the way the CTCs are collected?

Point 2:

Some statistical issues should be addressed.

The sample size is small for some of the groups studied (kidney, pancreas). Authors should recognize it and be cautious in interpreting these results and comparisons made.

Figure 4c and 4d show that a positive skewness exist on the data distribution. Authors use superposed red lines on data distribution to indicate mean and standard deviation, but it could be more informative to use classical box plots based on median, interquartile ranges, maximum and minimum whiskers, and outliers.

Regarding the bulk size distribution of the detected CTCs by each method (Figure 4e), do the authors verified the normality adjustment of the distribution? It seems to show positive skew (that is larger sizes), so the median instead of arithmetic mean could be a better centrality measurement and maybe the Welch corrected t-test could be unappropriated and an alternative non-parametric test could be a better option.

Point 3:

Independently of the comments made above, the discussion by the authors suggest that the apparent bias towards larger CTC sizes by the CTC-FIND method is due to a loss of small CTCs due to filtering. Do they have considered other methodological aspects that could give rise to this result?

Several differences in sample collection and pre-staining between the CTC-FIND and RareCyte methods could have an impact on this:

  • CTC-FIND method uses EDTA collection tubes and samples are processed within a 10h period while RareCyte method uses AccuCyte collection tubes (containing some kind of cell preservative, I suppose) and samples are processed within a 48h period
  • The CTC collection principles of each method are completely different
  • RareCyte method includes an air drying step and a -20ºC storage period before cell fixation and staining
  • CTC-FIND method uses 10’ rt formalin fixation, while RareCyte method uses 60’ rt formalin fixation after thawing

How these differences can influence cell morphology? Could CTCs shape and volume be affected by AccuCyte tube preservatives or extended period maintained at room temperature? Could the dielectrophoretic concentration procedure have a cell-size differential efficacy yielding a large CTC size enrichment compared to small CTCs? I consider especially critical the pre-fixation drying and freezing steps in the RareCyte method as far as dehydration caused by both processes can produce cell shrinkage. This could partially explain the apparently different size distribution obtained for each method as a methodological artefact.

Author Response

Dear Editors and Reviewer#1,

We sincerely thank the editor and reviewers for their careful consideration of the manuscript and for the thoughtful review and suggestions for revision that greatly improved the work. We have made substantial revisions to the manuscript at the reviewers’ suggestion and include a detailed point-by-point response below.

Comments from Reviewer#1, Point 1
According to figure 2b, for approximately two thirds of the samples the CTC-FIND counts are higher than RareCyte counts. How do the authors explain this if the RareCyte method as it is expected to be comprehensive in the way the CTCs are collected?

(Response to the reviewer, Point 1)
Thank you for your valuable comments. In this revision, we added explanation on page 9, line 305–311. Theoretically, the RareCyte system can comprehensively detect all the CTCs by collecting the buffy coat which contains all the nucleated cells in blood samples. However, in real practice, there are multiple steps that can cause cell loss, i.e. pipetting, tube switching, etc. Particularly, in the RareCyte system, a certain proportion of CTC loss could happen during the staining process on the slides. Instead, In the CTC-FIND system, CTCs were stained directly on the filter, which may prevent the CTC loss during staining since they could not pass through the filter pores.

Comments from Reviewer#1, Point 2
Some statistical issues should be addressed.

The sample size is small for some of the groups studied (kidney, pancreas). Authors should recognize it and be cautious in interpreting these results and comparisons made.

Figure 4c and 4d show that a positive skewness exist on the data distribution. Authors use superposed red lines on data distribution to indicate mean and standard deviation, but it could be more informative to use classical box plots based on median, interquartile ranges, maximum and minimum whiskers, and outliers.

Regarding the bulk size distribution of the detected CTCs by each method (Figure 4e), do the authors verified the normality adjustment of the distribution? It seems to show positive skew (that is larger sizes), so the median instead of arithmetic mean could be a better centrality measurement and maybe the Welch corrected t-test could be unappropriated and an alternative non-parametric test could be a better option.

(Response to the reviewer, Point 2)
Thank you for your valuable comments. In this revision, we revised Figure 4c and 4d (box plots based on the median). In addition, we use the median value for cell size distribution and we also revised explanation on page 1, abstract, line 26, on page 5, section of “2.4. Comparison of the CTC sizes between the two Methods” and on page 6, Figure 4 legend, line 221–226. The trend of the results did not change.
Normality for the data of each method was tested by Shapiro-Wilk test. They were not normally distributed (the CTC-FIND method, the RareCyte method, p<0.0001). For this reason, a non-parametric test (Mann Whitney test) was conducted for difference in the median cell size between two methods.

Comments from Reviewer#1, Point 3
Independently of the comments made above, the discussion by the authors suggest that the apparent bias towards larger CTC sizes by the CTC-FIND method is due to a loss of small CTCs due to filtering. Do they have considered other methodological aspects that could give rise to this result?

Several differences in sample collection and pre-staining between the CTC-FIND and RareCyte methods could have an impact on this:

CTC-FIND method uses EDTA collection tubes and samples are processed within a 10 h period while RareCyte method uses AccuCyte collection tubes (containing some kind of cell preservative, I suppose) and samples are processed within a 48 h period
The CTC collection principles of each method are completely different
RareCyte method includes an air drying step and a -20ºC storage period before cell fixation and staining
CTC-FIND method uses 10’ rt formalin fixation, while RareCyte method uses 60’ rt formalin fixation after thawing
How these differences can influence cell morphology? Could CTCs shape and volume be affected by AccuCyte tube preservatives or extended period maintained at room temperature? Could the dielectrophoretic concentration procedure have a cell-size differential efficacy yielding a large CTC size enrichment compared to small CTCs? I consider especially critical the pre-fixation drying and freezing steps in the RareCyte method as far as dehydration caused by both processes can produce cell shrinkage. This could partially explain the apparently different size distribution obtained for each method as a methodological artefact.

(Response to the reviewer, Point 3)
Thank you for your valuable comments. In this revision, we added explanation on page 9–10, line 320–340. We considered the difference of the staining processing might contribute to this result. In the RareCyte method, it is possible that large CTCs on the slide might have been washed away during the staining processing. There are multiple washing steps before or within the Autostainer. As result of these process, large CTCs on the slide might be easily washed away compared to small cells during the staining processing. This is because shear stress caused by the liquid flow affects large cells more strongly than small cells [1]. Another possibility is that large CTCs may be lost when the density of a large CTC (the specific gravity of a large CTCs) is slightly higher and the large CTCs may move to an erythrocyte layer in the AccuCyte tube after using a centrifuge. We also considered the difference of the blood collection tube. The EDTA collection tube does not make CTCs stable. In this study, after collecting blood, whole blood in the EDTA collection tube was processed via CTC-FIND as soon as possible.
 The AccuCyte collection tube makes CTCs stable for 72 hours (72 h) because the tube is included fixation reagent for CTCs. The fixation reagent also makes cells rigid (loss of deformability). These cells will clog the narrow slit-shaped filter, so that we did not use the tube for CTC-FIND.
According to the articles using the similar concept of the blood collection tube [2], Cellsave™ (Menarini), including fixation reagent, produced ~6% cell shrinkage during 48 h preservation. In general, the cells staying in EDTA for a long time might affect the cell size and fixation can cause some level of cell shrinkage. We need to think about this and further validation is needed.
In this study, we didn’t use the dielectrophoretic concentration. To analyze the CTC capture bias and detection capability of the slit filter isolation unit of CTC-FIND, we compared it to a selection-free method using the RareCyte system. For this reason, though CTC-FIND has a purification and enrichment unit, including the dielectrophoretic concentration, we did not use this unit. When it comes to the effects of dielectrophoretic concentration, large cells are affected more strongly by the dielectrophoretic force compared to small cells. Our dielectrophoretic concentration procedures are designed to trap cells on the bottom of the microfluidic channel in the dielectrophoretic device. For this reason, large cells are also affected by liquid flows (Shear stress). Trapped cell efficacy is related to the balance of the dielectrophoretic force with liquid flow in the dielectrophoretic device. This process of CTCFIND is designed to enrich large cells as well as small cells efficiently [1].

1. Kim, S.H.; Ito, H.; Kozuka, M.; Hirai, M.; Fujii, T. Localization of low-abundant cancer cells in a sharply expanded microfluidic step-channel using dielectrophoresis. Biomicrofluidics 2017, 11, doi:10.1063/1.4998756.
2. Park, S.; Ang, R.R.; Duffy, S.P.; Bazov, J.; Chi, K.N.; Black, P.C.; Ma, H. Morphological differences between circulating tumor cells from prostate cancer patients and cultured prostate cancer cells. PLoS One 2014, 9, e85264, doi:10.1371/journal.pone.0085264.

Again, we sincerely thank you for giving us the opportunity to improve our manuscript with your thoughtful review and suggestions. We have worked hard to incorporate your feedback and hope that these revisions persuade you to accept our submission.

Sincerely,

Reviewer 2 Report

In this article, the authors analysed the CTC capture bias and detection capability of the slit filter isolation unit of CTC-FIND, and they compared it to RareCyte method, a selection-free method, which collects the buffy coat, including CTCs and leukocytes, from whole blood.

In general, this is a good and interesting work. However, I have some comments and elucidations:

- Their report seems to suggest that both methods are useful to detect CTCs in liquid biopsy, however CTC-FIND method seems to find a higher number of CTCs than RareCyte one. In my opinion, it must discuss by what is due, such as type of cancers.

- The authors analysed CTCs in 36 samples: 21 are prostate cancer, 9 are bladder cancer and only 3 for kidney and pancreatic cancer. Prostate cancer contributes more than 50% to overall results. These are preliminary and exploratory results, for this reason it is mandatory, in my opinion, to consider other types of tumors in increasing the number of cases.

- Figure 5 d and e: (n=…) what refer to..? Number of CTCs analysed per slides?

- In the discussion the authors have to argument the results that they obtained about the phenotype. In the figure 5 d and e is showed the % of three different phenotype: which is the explanation for the absence of CK-EpCAM+ bladder and kidney CTCs detected by RareCyte method? I know that that they have only 3 pancreatic cases, but the results that they obtained is very different using two methods: CTC-FIND method detected only CK+/EpCAM- and CK+/EpCAM+, instead CTC-RareCyte detected also CK-/EpCAM+ (40%). Why this difference? RareCyte is more sensitive or not?  

Author Response

Dear Editors and Reviewer#2,

We sincerely thank the editor and reviewers for their careful consideration of the manuscript and for the thoughtful review and suggestions for revision that greatly improved the work. We have made substantial revisions to the manuscript at the reviewers’ suggestion and include a detailed point-by-point response below.

Comments from Reviewer#2, Q1
Their report seems to suggest that both methods are useful to detect CTCs in liquid biopsy, however CTC-FIND method seems to find a higher number of CTCs than RareCyte one. In my opinion, it must discuss by what is due, such as type of cancers.

(Response to the reviewer)
Thank you for your valuable comments. In this revision, we added explanation on page 9, line 305–311. Theoretically, the RareCyte system can comprehensively detect all the CTCs by collecting the buffy coat which contains all the nucleated cells in blood samples. However, in real practice, there are multiple steps that can cause cell loss, i.e. pipetting, tube switching, etc. Particularly, in the RareCyte system, a certain proportion of CTC loss could happen during the staining process on the slides. Instead, In the CTC-FIND system, CTCs were stained directly on the filter, which may prevent the CTC loss during staining since they could not pass through the filter pores.

Comments from Reviewer#2, Q2
The authors analyzed CTCs in 36 samples: 21 are prostate cancer, 9 are bladder cancer and only 3 for kidney and pancreatic cancer. Prostate cancer contributes more than 50% to overall results. These are preliminary and exploratory results, for this reason it is mandatory, in my opinion, to consider other types of tumors in increasing the number of cases.

(Response to the reviewer)
Thank you for your valuable comments. In this revision, we added explanation on page 11, line 390–392. We agree with you that prostate cancer CTCs contributed dominantly to overall results. These results may also be affected by the proportion of cases in each cancer type. These findings need to be further validated in other cancer types and a larger cohort size.

Comments from Reviewer#2, Q3
Figure 5 d and e: (n=…) what refer to..? Number of CTCs analyzed per slides?

(Response to the reviewer)
Thank you for your valuable comments. In this revision, we revised Figure 5d and 5e.
“n =” indicated total number of CTCs in each cancer type in this Figure. To avoid confusion, we changed “n =” to “total…cells”.

Comments from Reviewer#2, Q4
In the figure 5 d and e is showed the % of three different phenotype: which is the explanation for the absence of CK-EpCAM+ bladder and kidney CTCs detected by RareCyte method?

(Response to the reviewer)
Thank you for your valuable comments. In this revision, we added explanation on page 10, line 382–386. Both methods demonstrated that CK-EPCAM+ CTCs are very rare/absent in bladder cancer and kidney cancer samples. Even in CTCFIND, only 5.6% and 2.1% CTCs are CK-EPCAM+, which is 6 out of 106 and 1 out of 48. The difference between the two methods in this subtype is not statistically significant (Fisher's exact test, p>0.05).

Comments from Reviewer#2, Q5
CTC-FIND method detected only CK+/EpCAM- and CK+/EpCAM+, instead CTC-RareCyte detected also CK-/EpCAM+ (40%). Why this difference? RareCyte is more sensitive or not?

(Response to the reviewer)
Thank you for your valuable comment. In this revision, we added explanation on page 10, line 386–389. This difference can be due to the limited number of CTCs captured by either method in pancreatic cancer patients. From 3 pancreatic cancer blood samples, a total of 4 CTCs were yielded by the CTC-FIND method and a total of 5 CTCs were yielded via the RareCyte method.
For CTC phenotype rate, In the CTCFIND method, 3 CK+/EpCAM- CTCs (75%), one CK-/EpCAM+ CTCs (25%). In the RareCyte method hand, 2 CK+/EpCAM- CTCs (40%), one CK-/EpCAM+ (20%), 2 CK+/EpCAM+ CTCs (40%)

Again, we sincerely thank you for giving us the opportunity to improve our manuscript with your thoughtful review and suggestions. We have worked hard to incorporate your feedback and hope that these revisions persuade you to accept our submission.

Sincerely,

Reviewer 3 Report

The study presents a head-to-head comparison of two CTC isolation methods which may also detect EPCAM negative cells. It is well designed and performed with blood samples from patients with metastatic disease.

Minor comments
- The CTC-FIND method captures cancer cells due to bigger size and lower deformability than those of leukocytes. It might be of interest how many normal blood cells are retained on the CTC-FIND filter.
- Does the CTC-FIND instrument enables the release of the CTCs (before formalin fixation) for in-vitro culture or xenografting?
- The authors properly discuss a higher count of small CTCs detected by AccuCyte in comparison to CTC-FIND. Why the AccuCyte system less frequently detects larger cells in comparison to CTC-FIND (Figure 4e)? Can the cells be enlarged by deformation when captured on the slit pore in the CTC-FIND system?

Author Response

Dear Editors and Reviewer#3,

We sincerely thank the editor and reviewers for their careful consideration of the manuscript and for the thoughtful review and suggestions for revision that greatly improved the work. We have made substantial revisions to the manuscript at the reviewers’ suggestion and include a detailed point-by-point response below.

Comments from Reviewer#3
The study presents a head-to-head comparison of two CTC isolation methods which may also detect EPCAM negative cells. It is well designed and performed with blood samples from patients with metastatic disease.
Minor comments

Comments from Reviewer#3, Q1
The CTC-FIND method captures cancer cells due to bigger size and lower deformability than those of leukocytes. It might be of interest how many normal blood cells are retained on the CTC-FIND filter.

(Response to the reviewer, Q1)
Thank you for your valuable comments. In this study, CyteFinder was not able to count the total cell numbers on each slide. In our previous study (spike-in experiment), the number of leukocytes on each filter ranged from 10,000 cells to 40,000 cells when filtering 7.5 mL of healthy whole blood.

Comments from Reviewer#3, Q2
Does the CTC-FIND instrument enables the release of the CTCs (before formalin fixation) for in-vitro culture or xenografting?

(Response to the reviewer, Q2)
Thank you for your valuable comments. In a spike-in experiment, cancer cells (before formalin-fixed) were able to be cultured after being collected from the filter. Theoretically, CTC-FIND potentially provides live CTCs for in-vitro culture or xenografting. However, we have never used clinical specimens to verify that CTC-FIND can provide CTCs for such applications.

Comments from Reviewer#3, Q3
The authors properly discuss a higher count of small CTCs detected by AccuCyte in comparison to CTC-FIND. Why the AccuCyte system less frequently detects larger cells in comparison to CTC-FIND (Figure 4e)?

(Response to the reviewer, Q3)
Thank you for your valuable comments. In this revision, we added explanation on page 9, line 320–327. We considered that the loss of some large CTCs could happen during the staining process in the RareCyte method. There are multiple washing steps before or within the Autostainer. As result of these process, large CTCs on the slide might be easily washed away compared to small cells during the staining processing. This is because shear stress caused by the liquid flow affects large cells more strongly than small cells [1]. However, in CTCFIND system, all the cells were stained on the filter, which will avoid the same type of cell loss. Another possibility is that large CTCs may be lost when the density of a large CTC (the specific gravity of a large CTC) is slightly higher and the large CTCs may move to an erythrocyte layer in the AccuCyte tube after using a centrifuge.

1. Kim, S.H.; Ito, H.; Kozuka, M.; Hirai, M.; Fujii, T. Localization of low-abundant cancer cells in a sharply expanded microfluidic step-channel using dielectrophoresis. Biomicrofluidics 2017, 11, doi:10.1063/1.4998756.

Comments from Reviewer#3, Q4
Can the cells be enlarged by deformation when captured on the slit pore in the CTC-FIND system?

(Response to the reviewer, Q4)
Thank you for your valuable thought and question. Unfortunately, we don’t have any live video or comparison with other non-slit filter devices to prove whether or not the cells will be enlarged by deformation. However, theoretically, we think the cells may be shrinking because cells are affected by water pressure from all directions and also cells are drawn into the pores via flows and the pressure drop (∆P) across the filter. Another possibility is that we consider that cells will recover original size by their viscoelasticity because the pressure drop (∆P) and flow towards the filter pores will be released after each process in the CTC-FIND method.

Again, we sincerely thank you for giving us the opportunity to improve our manuscript with your thoughtful review and suggestions. We have worked hard to incorporate your feedback and hope that these revisions persuade you to accept our submission.

Sincerely,

Round 2

Reviewer 2 Report

Thanks to the authors for the replay. 

Despite a no-correspondance with pages and lines reported in the rebutal, I agree with the modifications that I find in the revised manuscritp.